# Efficacy of Four Insecticides Applied to Fortified Rice with Basil against Major Stored-Product Insect Species

Paraskevi Agrafioti *, Mariastela Vrontaki, Evagelia Lampiri, Christos I. Rumbos and Christos G. Athanassiou

Laboratory of Entomology and Agricultural Zoology, Department of Agriculture, Crop Production and Rural Environment, University of Thessaly, Phytokou Str., 38446 Volos, Greece; mariastelaav@gmail.com (M.V.); elampiri@uth.gr (E.L.); crumbos@uth.gr (C.I.R.); athanassiou@uth.gr (C.G.A.)
* Correspondence: agrafiot@agr.uth.gr; Tel.: +30-2421093109

**Abstract:** Rice is one of the most important foods since it is grown in many countries and consumed by the majority of the population. Ensuring food security through the protection of stored-product commodities has become one of the most important priorities worldwide. In this study, the effects of three insecticidal formulations and an available diatomaceous earth formulation on basil-fortified rice against the major stored-product insects were evaluated. The label dose of each insecticidal formulation was used. Insect mortality was determined after 1, 3, 7, 14 and 21 days for each species. The entire experiment was repeated three times by preparing different lots of treated and untreated rice for each treatment. The results of the diatomaceous earth treatments showed that the most susceptible individuals were *Sitophilus oryzae* adults and *Oryzaephilus surinamensis* larvae, while the least susceptible individuals were *Rhyzopertha dominica* and *Tribolium castaneum* adults. The tested insecticides were effective against *S. oryzae*, as mortality was 100%. Between the two pyrethroids, deltamethrin was more effective than cypermethrin in the tested insecticides. Our findings indicate that there are available insecticides on the market which can be obtained successfully for the durable protection of agricultural commodities after the harvest stage.

**Keywords:** fortification; rice; insecticides; diatomaceous earth; stored-product insects; mortality

## 1. Introduction

Rice [*Oryza sativa* L. (Poales: Poaceae)] is considered one of the most important foods worldwide, as it is cultivated in many countries and continuously consumed by the vast majority of the human population [1]. In more than 100 countries in the world, especially in Asia, sub-Saharan Africa and South America, rice is the main staple food [2–4]. Specifically, it has been assessed that 90% of rice is produced and consumed in the Asia–Pacific region [2,3]. Moreover, the rice production of China and India alone accounts for approximately 50% of the world's rice production [2]. Other rice-producing countries are Brazil, the United States, Egypt, Madagascar and Nigeria, accounting for 5% of the world's rice production [5]. Given that the global consumption and production of milled rice are rapidly increasing, the post-harvest safety of rice production is crucial.

Ensuring food security through the protection of stored commodities has become one of the most important priorities for both developed and developing countries worldwide [6]. The most effective method for the control of stored-product insects is the use of chemical insecticides, namely contact insecticides and fumigants [6]. Phosphine is the main gas for the disinfestation of the most durable commodities, such as cereals, dried fruits and tobacco, in different storage facilities globally [7–9]. Nevertheless, the improper use (leaky structures, low concentration, no fumigation plan) of this insecticide has resulted in the development of phosphine resistance in most stored-product insect species [10–13].

Several papers have been published on the efficacy of the contact insecticides, such as organophosphorus compounds (OPs) and synthetic pyrethroids, that are wielded for the

post-harvest protection of agricultural commodities [14–16]. For instance, cypermethrin and deltamethrin are two broad-spectrum contact insecticides that are commonly used against insects of public-health importance [17–19]. Cypermethrin is a mixture of four cis- and four trans-isomers, two of which show insecticidal activity [20,21]. It has been examined against different insect pests, such as cockroaches [22], mosquitoes [19] and crop insect species [23,24]. There are also several studies that have tested the efficacy of cypermethrin after direct application on grains against stored-product insects [16,25]. Deltamethrin is also used for the control of stored-product beetles [15,26,27]. For instance, Kavallieratos et al. [15] tested the efficacy of deltamethrin against stored-product insects on partially treated rice and reported low immediate and delayed mortality after 24 h of exposure. Regarding OPs, pirimiphos-methyl is considered the most commonly used grain protectant globally. More recently, studies have shown that pirimiphos-methyl is extremely efficient when it is sprayed directly on grains [28,29]. For example, Rumbos et al. [29] found that *Sitophilus* species were highly susceptible to pirimiphos-methyl formulations, since 100% mortality was noted after 7 d of exposure.

Inert dusts have been extensively studied for several durable agricultural commodities after the harvest stage [30]. The most well known inert material that has been successfully used and is currently registered globally is diatomaceous earth (DE) [31,32]. Several studies have been carried out in order to evaluate the effect of various types of DEs in stored-product pest management [16]. Nevertheless, the effectiveness of DEs can be influenced extremely depending on the concentration, time interval, temperature, relative humidity, commodity and different life stages [16,31–33]. For example, Baliota et al. [16] examined the influence of temperature on the insecticidal effect of diatomaceous earth against the rusty grain beetle, *Cryptolestes ferruguneus* Stephens (Coleoptera: Laemophloeideae), the sawtoothed grain beetle, *Oryzaephilus surinamensis* (L.) (Coleoptera: Silvanidae), and the confused flour beetle, *Tribolium confusum* Jacquelin du Val (Coleoptera: Tenebrionidae), and they clearly found that the increase in temperature to 30 °C and the highest dose rate increased the adult mortality levels.

The consumption of fortified staple foods has been proven to increase vitamin and mineral intake [34–37]. Many published papers have shown that the fortification of rice in plant extracts has the ability to absorb and maintain a significant amount of antioxidants, such as color and aromas [36,37]. Specifically, Igoumenidis et al. [36] showed that rice fortification with herbs could be utilized for the production of quick-cooking fortified rice, and the final product contains better nutritional attributes than cooked white milled rice. On the other hand, Agrafioti et al. [38] evaluated the stored-product insects in fortified rice with herbs (spearmint and basil) and highlighted that the fortification increased the mortality rate and suspended the offspring production. However, no information was available to determine the insecticidal effectiveness of different insecticides through the chemical and physical modes of action. To close this gap, three insecticides and an available diatomaceous earth formulation were sprayed and applied on fortified rice for protection against stored-product insect species. For this purpose, and considering that there are no relevant available data, a series of bioassays was carried out to comparatively examine the effectiveness of the different insecticides against major storage-product insect species. Thus, for the experiment, two primary colonizers were chosen, the lesser grain borer, *Rhyzopertha dominica* (F.) (Coleoptera: Bostrychidae), and the rice weevil, *Sitophilus oryzae* (L.) (Coleoptera: Curculionidae), as well as two secondary colonizers, i.e., the red flour beetle, *Tribolium castaneum* (Herbst) (Coleoptera: Tenebrionidae), and *O. surinamensis*.

## 2. Materials and Methods

### 2.1. Tested Insects

Adults of *S. oryzae*, *R. dominica*, *T. castaneum* and *O. surinamensis*, as well as larvae of *T. castaneum* and *O. surinamenis*, were utilized in the bioassays. The tested species were kept at the Laboratory of Entomology and Agricultural Zoology (LEAZ), Department of Agriculture, Crop Production and Rural Environment, University of Thessaly, at 25 °C

and 65% relative humidity (r.h.) with full obscurity. Whole wheat kernels were used as feeding substrate for *S. oryzae* and *R. dominica,* whereas *T. castaneum* and *O. surinamensis* were fed wheat flour and quakers, respectively. Adult individuals less than a month old and last-instar larvae were tested in the bioassays. All insect cultures have been maintained under controlled conditions for more than 20 years.

### 2.2. Commodity and Insecticide Treatments

In this study, insect- and insecticide-free basil-fortified rice was the commodity used, hereafter referred to as rice. The procedure for the rice fortification was previously described based on references [36,38]. Prior to the experiment, fortified rice was left in a controlled condition ($-20\ ^{\circ}$C) for more than two weeks.

Three commercial insecticidal formulations were assessed in the bioassays: deltamethrin (K-Othrine® 250 SC, containing 25% of deltamethrin, Bayer Hellas, Athens, Greece), cypermethrin (Talisma EC, containing 8% of cypermethrin, Agriphar S.A. Hellas, Athens, Greece) and pirimiphos-methyl (Actellic® 50 EC, containing 50% of pirimiphos-methyl, Syngenta Hellas, Athens, Greece). The diatomaceous earth Silicid (containing >99% of $SiO_2$, BioFa GmbH, Munchigen, Germany) was used in the experiments.

### 2.3. Insecticidal Efficacy

The label dose for each of the aforementioned contact insecticide was used. For cypermethrin, 0.02 mL of the formulation per kg of grain was used; for deltamethrin, 9.7 mg of grain/$m^2$ was applied; and for pirimiphos-methyl, 500 g/L of the active ingredient was used as suggested by Rumbos et al. [29]. The diatomaceous earth formulation was applied at 1 g/kg (1000 ppm). Insecticide spraying solutions were created by diluting the specific quantity of each insecticidal formulation, i.e., 1.1, 8.25 and 5.5 mL for cypermethrin, deltamethrin and pirimiphos-methyl, respectively, in 50 mL of distilled water. A total of 1 mL of the spraying solution was applied on a specific amount of rice (110 g) using the airbrush (Badger 100, Kyoto BD-183 K Graphotech, Japan). An extra series of rice batches was sprayed with distilled water, serving as a control. Then, the treated commodity was transferred in glass jars and shaken manually (2 min) to achieve equal distribution of the insecticide/dust formulation throughout the commodity. Two grams (2 g) of commodity samples from each treatment and twenty adults of each insect species were allocated in cylindrical plastic vials (Rotilabo®-sample tins with snap-on lids, 3 cm in diameter, 8 cm in height, Carl Roth Gmbh & Co., Kg, Karlsruhe, Germany), with different sets of vials for each combination (insect species × insecticidal formulation). Furthermore, the top quarter of the vials was covered with Fluon (polytetrafluoroethylene, Northern Products, Woonsocket, RI, USA) to prevent insects from escaping. The whole bioassay was carried out three times (three series of vials) by preparing different lots of treated and untreated rice each time (3 × 3 vials for each treatment). The vials were kept in controlled conditions at 25 $^{\circ}$C, 55% r.h., with full of darkness. Individual mortality was counted after 1, 3, 7, 14 and 21 days for each species.

### 2.4. Statistical Analysis

Prior to analysis, all data were tested for normality and homogeneity using Kolmogorov–Smirnov and Levene's tests, respectively. When the equality criteria were not met, the O'Brien or Brown–Forsythe tests were utilized. When the Levene, O'Brien or Brown–Forsythe tests were not equal, data were transformed (Exponential, Log10, Squish, Factional, Root, Gamma or LGamma) to meet the criteria. Control mortality for diatomaceous earth formulation was extremely low; thus, the data were excluded from the analysis. In the case of diatomaceous earth formulation, one-way ANOVA was utilized to determine the differences among the exposure intervals (1, 3, 7, 14 and 21 d) for each tested insect species. For the mortality data, two-way ANOVA was performed with exposure interval as a repeated variable and the insecticide active ingredient (cypermethrin, deltamethrin, pirimiphos-methyl and control) as main effects. In order to determine if there were differ-

ences between the means of insect mortality for each active ingredient within each exposure time, comparison of means was performed using the Tukey–Kramer HSD test at the 5% level [39].

## 3. Results

### 3.1. Insect Mortality

In most of the cases, for all examined species, the main effects (treatment, time) and the associated interaction (treatment $\times$ time) were significant with $p < 0.001$ (Tables 1 and 2).

**Table 1.** One-way ANOVA parameters for the individuals for main effect for mortality levels of *Rhyzopertha dominica*, *Sitophilus oryzae*, *Tribolium castaneum* and *Oryzaephilus surinamensis* on rice treated with diatomaceous earth formulation at different exposure intervals.

| | | Adults | | | | | | | | Larvae | | | |
|---|---|---|---|---|---|---|---|---|---|---|---|---|---|
| **Source** | | *Rhyzopertha dominica* | | *Sitophilus oryzae* | | *Tribolium castaneum* | | *Oryzaephilus surinamensis* | | *Tribolium castaneum* | | *Oryzaephilus surinamensis* | |
| | ***df*** | **F** | ***p*** | **F** | ***p*** | **F** | ***p*** | **F** | ***p*** | **F** | ***p*** | **F** | ***p*** |
| Intercept | 1 | 8.49 | 0.019 | 0 | 1.0 | 19.69 | 0.002 | 947.76 | <0.001 | 42.91 | <0.001 | 10,488.2 | <0.001 |
| Time | 4 | 3.07 | 0.124 | 0 | 1.0 | 3.11 | 0.122 | 410.25 | <0.001 | 45.97 | <0.001 | 75.03 | <0.001 |

**Table 2.** Two-way ANOVA parameters for the individuals for main effects and associated interactions for mortality levels of *Rhyzopertha dominica*, *Sitophilus oryzae*, *Tribolium castaneum* and *Oryzaephilus surinamensis* on rice treated with cypermethrin, detlamehtrin, pirimiphos methyl and control at different exposure intervals.

| | | Adults | | | | | | | | Larvae | | | |
|---|---|---|---|---|---|---|---|---|---|---|---|---|---|
| **Source** | | *Rhyzopertha dominica* | | *Sitophilus oryzae* | | *Tribolium castaneum* | | *Oryzaephilus surinamensis* | | *Tribolium castaneum* | | *Oryzaephilus surinamensis* | |
| | ***df*** | **F** | ***p*** | **F** | ***p*** | **F** | ***p*** | **F** | ***p*** | **F** | ***p*** | **F** | ***p*** |
| Intercept | 1 | 2484.79 | <0.001 | 52,153.60 | <0.001 | 913.66 | <0.001 | 8102.71 | <0.001 | 287.59 | <0.001 | 11,634.67 | <0.001 |
| Treatment [a] | 3 | 461.30 | <0.001 | 3578.14 | <0.001 | 242.46 | <0.001 | 598.79 | <0.001 | 21.58 | <0.001 | 232.45 | <0.001 |
| Time | 4 | 747.87 | <0.001 | 24,063.47 | <0.001 | 316.07 | <0.001 | 3143.82 | <0.001 | 135.21 | <0.001 | 28,732.96 | <0.001 |
| Time $\times$ Treatment | 12 | 31.73 * | <0.001 | 84.50 * | <0.001 | 32.38 * | <0.001 | 54.41 * | <0.001 | 11.33 * | <0.001 | 64.12 * | <0.001 |

[a] Included three insecticides plus control. * Wilks's Lambda approximate.

### 3.2. Rhyzopertha dominica

Generally, low mortality (<20%) was detected for *R. dominica* adults exposed to fortified rice treated with Silicid (Figure 1). However, when fortified rice was treated with the different insecticidal formulations tested (cypermethrin, deltamethrin and pirimiphos-methyl), mortality was notably high (Figure 2). Specifically, after 3 d of exposure, complete control (100% mortality) was noted in rice treated with deltamethrin, significantly different to the rest of the formulations tested. After 7 d of exposure, the mortality levels were 98, 100 and 22% in rice treated with cypermethrin, deltamethrin and pirimiphos-methyl, respectively. Furthermore, after two weeks of exposure, complete control was achieved in rice treated with cypermethrin and deltamethrin. Finally, after 21 d of exposure, the lowest mortality levels were assessed for rice treated with pirimiphos-methyl; they did not exceed 30% (Figure 2). With the exception of Day 1, in the majority of the tests, significant differences in mortality rates were noted among the tested insecticides (Figure 2).

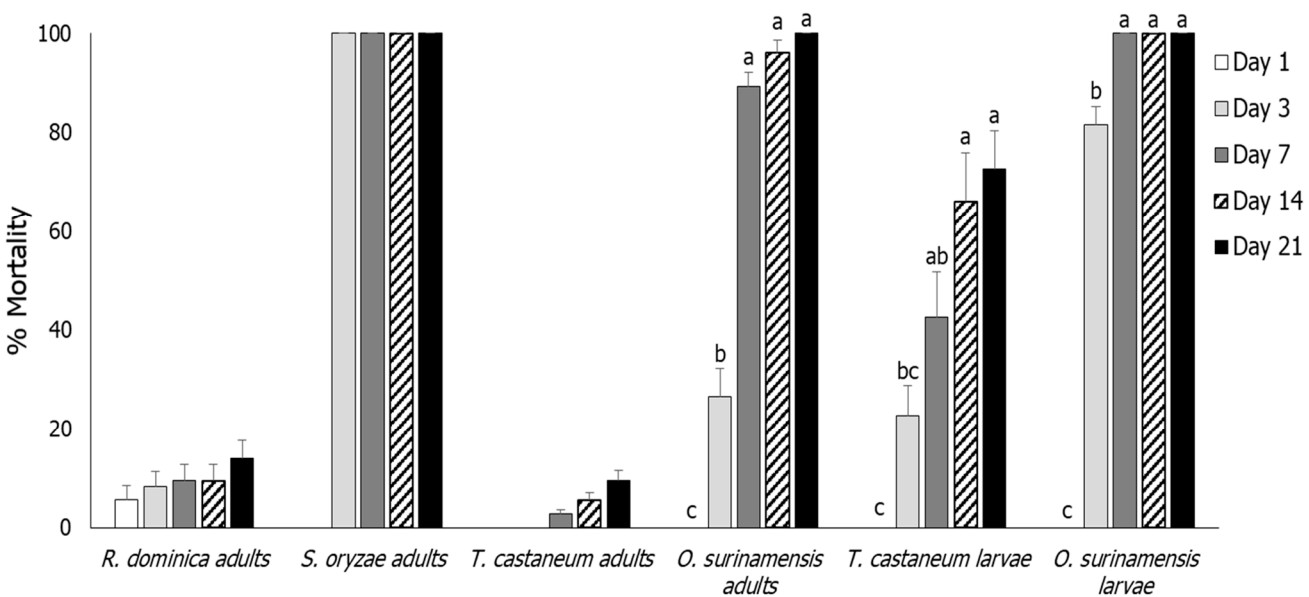

**Figure 1.** Mortality means (% ± SE) of adults of *Rhyzopertha dominica*, *Sitophilus oryzae*, *Tribolium castaneum* and *Oryzaephilus surinamensis* and larvae of *Tribolium castaneum* and *Oryzephilus surinamensis* after 1, 3, 7, 14 and 21 d of exposure to rice treated with a commercial diatomaceous earth formulation at 1000 ppm. For each insect species, means followed by the same lowercase letter do not differ significantly according to the Tukey–Kramer HSD test at $p < 0.05$. Where no letters exist, no significant differences were noted. ANOVA parameters for *T. castaneum* adults were F = 9.685 and $p < 0.001$; for *T. castaneum* larvae, they were F = 16.532 and $p < 0.001$. ANOVA parameters for *O. surinamensis* were F = 223.0 and $p < 0.001$ and for *O. surinamensis* larvae they were F = 678.20 and $p < 0.001$.

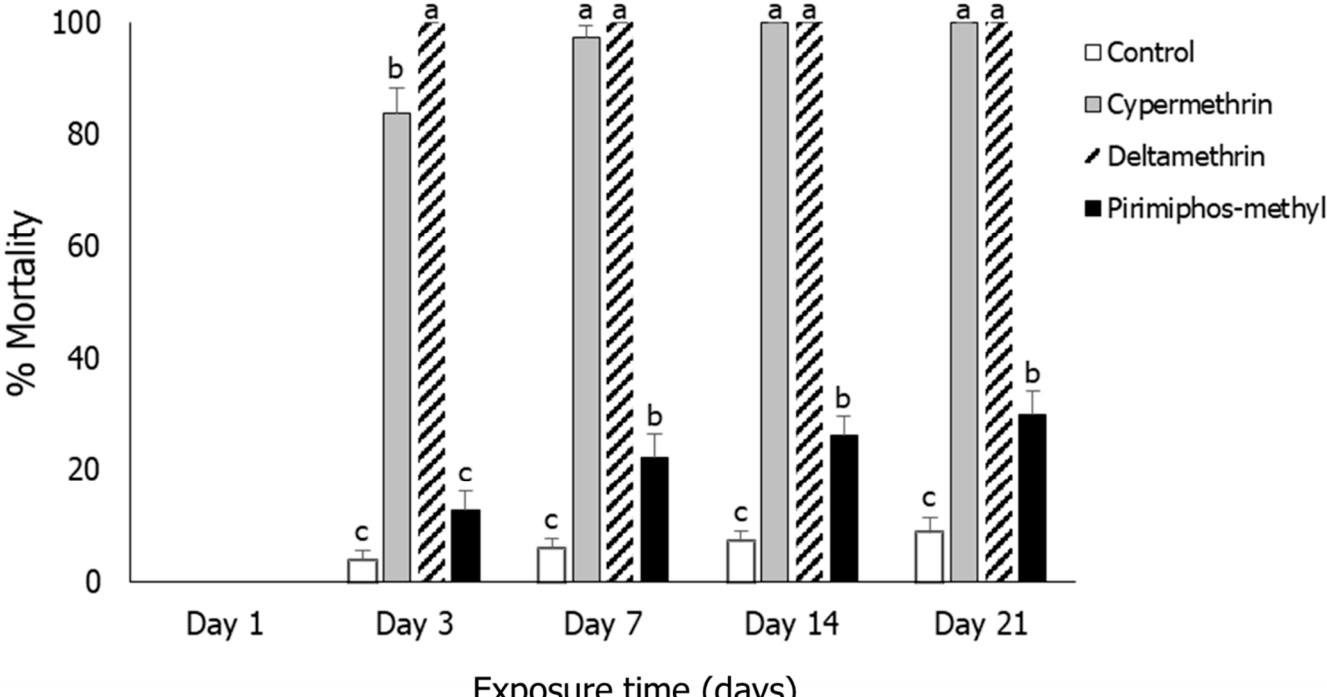

**Figure 2.** Mortality means (% ± SE) of *Rhyzopertha dominica* adults after exposure for 1, 3, 7, 14 and 21 days to rice treated with three insecticides (cypermethrin, deltamethrin, pirimiphos-methyl and control). Within each exposure interval, means followed by the same lowercase letter did not differ significantly according to the Tukey–Kramer HSD test at $p < 0.05$. Where no letters exist, no significant

differences were noted. ANOVA parameters: for Day 3, F = 282.48 and $p$ <0.001; for Day 7, F = 367.04 and $p$ < 0.001; for Day 14, F = 572.47 and $p$ < 0.001; and for Day 21, F = 360.19 and $p$ < 0.001. On Day 3, data were root-transformed and the O'Brien test was utilized (F = 2.77, $p$ < 0.057); on Day 7, data were squish-transformed and Brown–Forsythe was utilized (F = 2.38, $p$ < = 0.087); on Day 14, data were also squish-transformed and Brown–Forsythe was utilized (F = 2.44, $p$ < = 0.081); on Day 21, data were squish-transformed and Brown–Forsythe was utilized (F = 2.36, $p$ < = 0.089).

### 3.3. *Sitophilus oryzae*

Complete control was recorded for *S. oryzae* adults in rice treated with the diatomaceous earth formulation after one day of exposure (Figure 1). Nevertheless, at the same exposure interval, when rice was treated with the contact insecticides, zero mortality was observed (Figure 3). Interestingly, after 3 d of exposure, the mortality rates were increased for all tested insecticidal formulations (Figure 3). For all treatments, all insects were dead after 7 d of exposure (Figure 3).

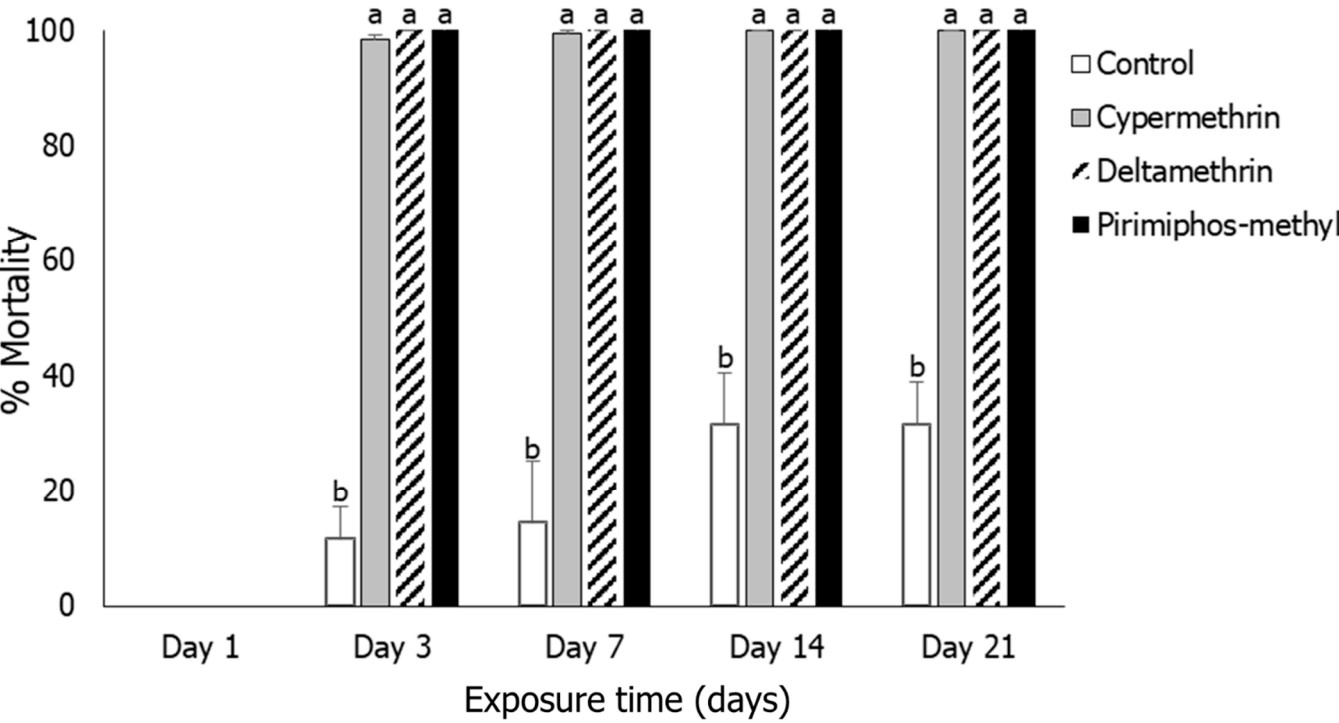

**Figure 3.** Mortality means (% ± SE) of *Sitophilus oryzae* adults after exposure for 1, 3, 7, 14 and 21 days to rice treated with three insecticides (cypermethrin, deltamethrin, pirimiphos-methyl and control). Within each exposure interval, means followed by the same lowercase letter did not differ significantly according to the Tukey–Kramer HSD test at $p$ < 0.05. Where no letters exist, no significant differences were noted. ANOVA parameters were noted for Day 3 (F = 62.78, $p$ < 0.001); for Day 7 (F = 62.88, $p$ < 0.001); for Day 14 (F = 59.50, $p$ < 0.001) and for Day 21 (F = 51.59, $p$ < 0.001). On Day 3, the Brown–Forsythe test was utilized ($p$ = 0.394); on Day 7, the O'Brien test was utilized ($p$ = 0.339); on Day 14, the O'Brien test was utilized ($p$ = 0.280); on Day 21, the O'Brien test was utilized ($p$ = 0.159).

### 3.4. *Tribolium castaneum*

Generally, extremely low mortality rates were found for *T. castaneum* adults when the rice was treated with the diatomaceous earth formulation (Figure 1). Indicatively, the highest mortality levels were recorded after 21 d of exposure and they did not exceed 10% (Figure 1). Low mortality rates were also recorded for cypermethrin after 3 d of exposure, in contrast with the other two insecticidal formulations (Figure 4). At longer exposure intervals (i.e., 7, 14 and 21 d of exposure), mortality increased in rice treated with deltamethrin and

pirimiphos-methyl. For instance, after 7 d of exposure, the highest mortality level was noted for deltamethrin, exceeding 84%, which was found to be significantly different to the relevant figures for the pirimiphos-methyl and cypermethrin (Figure 4). Moreover, significant differences were recorded among the examined insecticidal formulations after two weeks (Figure 4). Finally, complete control was achieved against *T. castaneum* adults in rice treated with deltamethrin after 21 d of exposure (Figure 4).

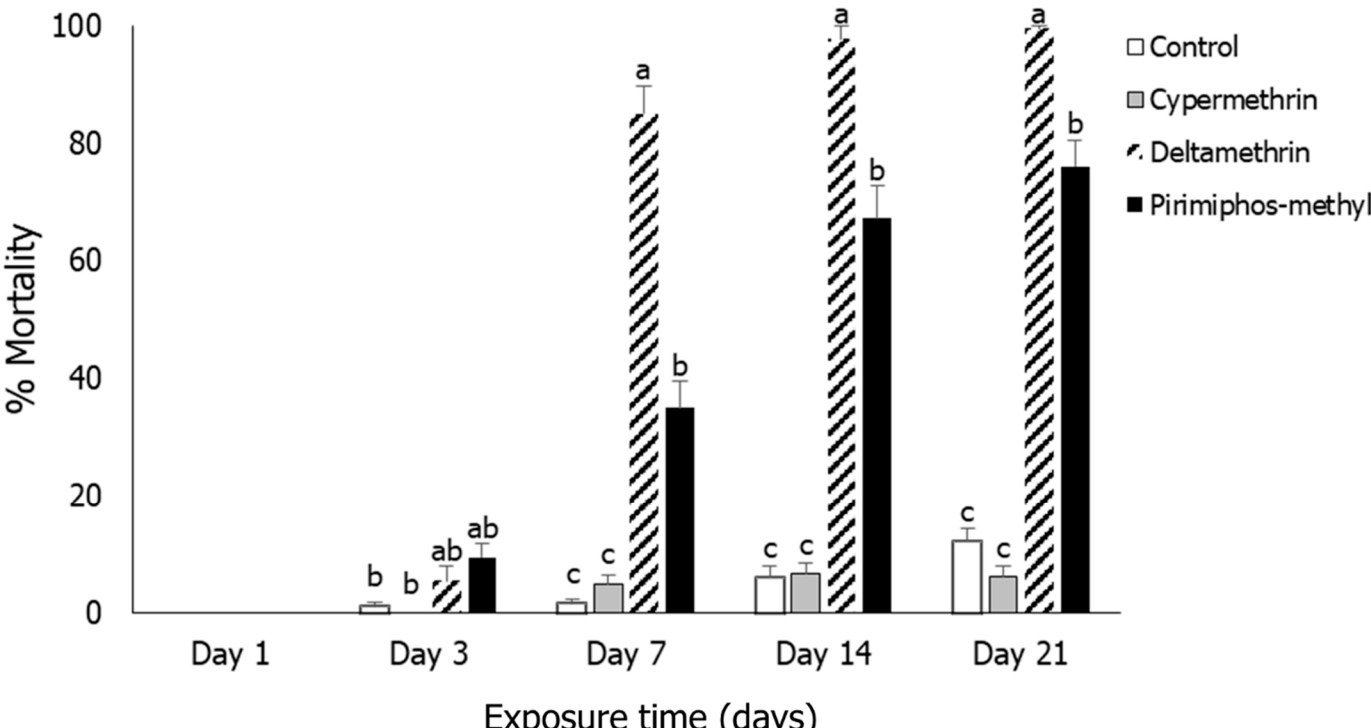

**Figure 4.** Mortality means (% $\pm$ SE) of *Tribolium castaneum* adults after exposure for 1, 3, 7, 14 and 21 days to rice treated with three insecticides (cypermethrin, deltamethrin, pirimiphos-methyl and control). Within each exposure interval, means followed by the same lowercase letter did not differ significantly according to the Tukey–Kramer HSD test at $p < 0.05$. Where no letters exists, no significant differences were noted. ANOVA parameters were noted for Day 3 (F = 6.13, $p$ =0.002); for Day 7 (F = 129.93, $p < 0.001$); for Day 14 (F = 197.03, $p < 0.001$) and for Day 21 (F = 295.01, $p < 0.001$). On Day 3, data were squish-transformed and the Brown–Forsythe test was utilized (F = 1.78, $p$ = 0.169); on Day 7, data were Log10-transformed and the O' Brien test was utilized (F = 1.96, $p$ = 0.147); on Day 14, the Brown–Forsythe test was utilized ($p$ = 0.058); and on Day 21, data were exp. transformed and the O'Brien test was utilized (F = 1.137, $p$ = 0.248).

Regarding *T. castaneum* larvae, increasing mortality levels were recorded after the application of Silicid with the increase in the exposure intervals (Figure 1). For instance, after 1, 3, 7, 14 and 21 d of exposure, the mortality levels were 0, 23, 43, 66 and 73% in the Silicid-treated rice (Figure 1). Similarly, in the vast majority of the tests, efficient control was indicated in the rice treated with cypermethrin and deltamethrin after 21 d of exposure (Figure 5). Indicatively, after 1 d of exposure, zero mortality was noted; however, after 3 d of exposure, the highest mortality was found in rice treated with cypermethrin (Figure 5). After 7 d of exposure, mortality was generally low in pirimiphos-methyl-treated rice, not exceeding 20%, and it was found to be significantly lower than the respective rates for deltamethrin and cypermethrin. The same pattern was found after two weeks (14 d of exposure) (Figure 5). Finally, after 21 d of exposure, deltamethrin and cypermethrin caused 86 and 98% mortality, respectively, significantly higher than pirimiphos-methyl (Figure 5).

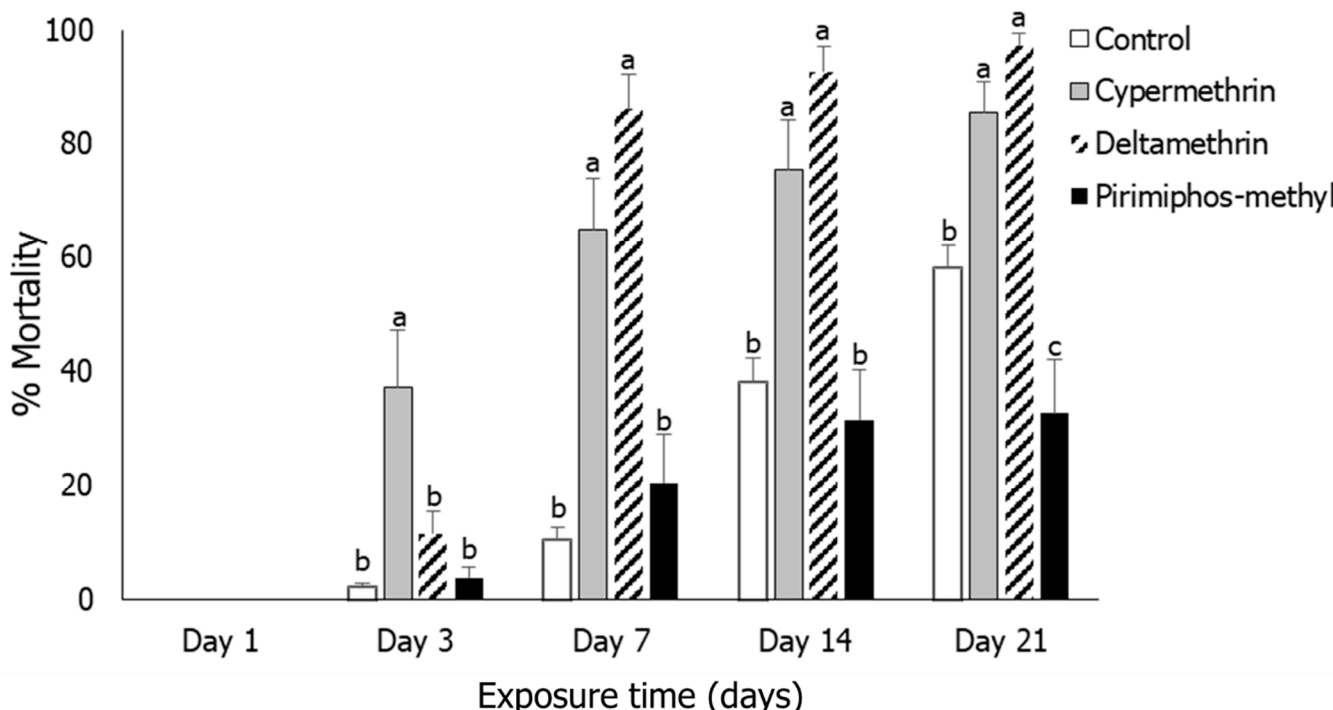

**Figure 5.** Mortality means (% ± SE) of *Tribolium castaneum* larvae after exposure for 1, 3, 7, 14 and 21 days to rice treated with three insecticides (cypermethrin, deltamethrin, pirimiphos-methyl and control). Within each exposure interval, means followed by the same lowercase letter did not differ significantly according to the Tukey–Kramer HSD test at $p < 0.05$. Where no letters exist, no significant differences were noted. ANOVA parameters were noted for Day 3 (F = 8.33, $p < 0.001$); for Day 7 (F = 26.02, $p < 0.001$); for Day 14 (F = 17.64, $p < 0.001$) and for Day 21 (F = 23.29, $p < 0.001$). On Day 3, data were exp. transformed and the O'Brien test was utilized (F = 1.13, $p = 0.348$); on Day 7, the Brown–Forsythe test was utilized ($p = 0.342$); on Day 14, the Brown–Forsythe test was utilized ($p = 0.344$); and on Day 21, the Brown–Forsythe test was utilized ($p = 0.132$).

*3.5. Oryzaephilus surinamensis*

In all cases, when the different insecticides and the diatomaceous earth formulation were applied to fortified rice on Day 1, mortality levels of *O. surinamensis* adults were zero (Figures 1, 6 and 7). After diatomaceous earth application, mortality reached 0, 27, 90, 96 and 100 % after 1, 3, 7, 14 and 21 d of exposure, respectively (Figure 1). The mortality levels in the treatment with insecticides were affected by the exposure time; however, mortality was high even after 3 d of exposure (Figure 6). For the same exposure interval, statistically significant differences were achieved among the tested insecticides, with the highest mortality (100%) achieved in deltamethrin-treated rice (Figure 6). After 7 d of exposure, 100% mortality was also noted for cypermethrin and deltamethrin-treated rice. Similar results were found after 14 d of exposure (Figure 6); no significant differences were recorded among the tested insecticides (Figure 6).

Regarding *O. surinamensis* larvae, mortality in diatomaceous-earth-treated rice achieved 82% after only 3 d of exposure, while 100% mortality was noted after 7 d of exposure (Figure 1). Similarly, when the tested insecticidal formulations were applied to rice, the larval stage was highly susceptible, since 100% mortality was highlighted for all insecticides after only 3 d of exposure (Figure 7).

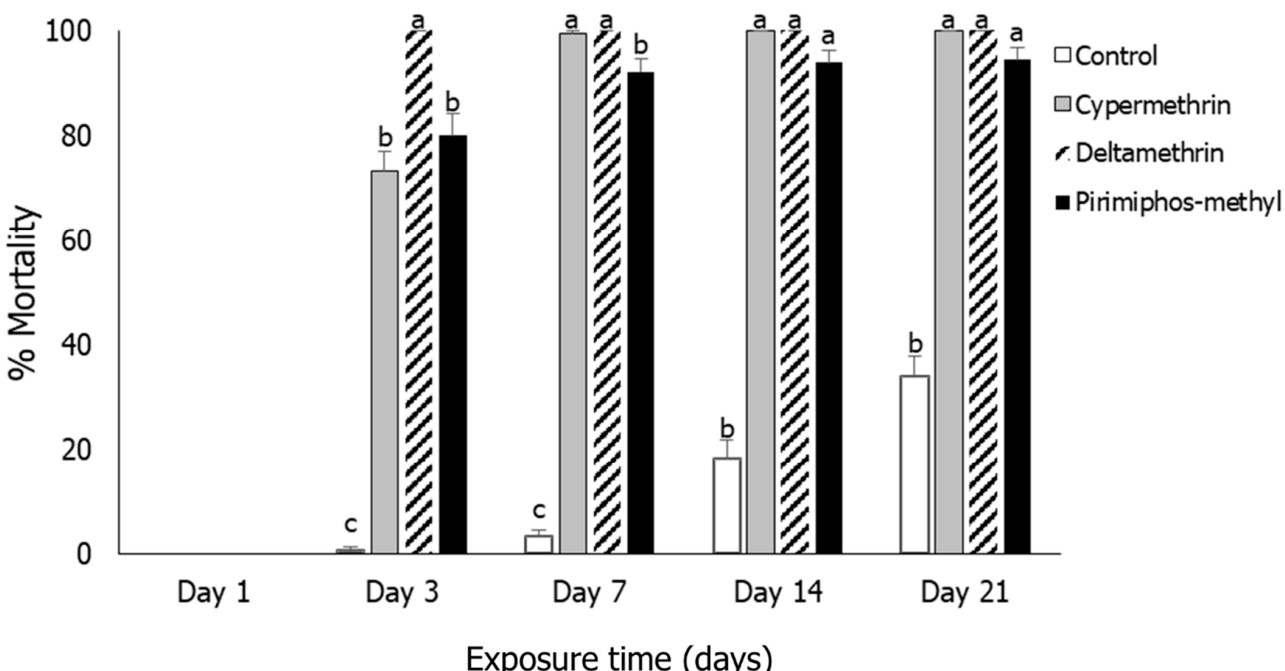

**Figure 6.** Mortality means (% ± SE) of *Oryzaephilus surinamensis* adults after exposure for 1, 3, 7, 14 and 21 days to rice treated with three insecticides (cypermethrin, deltamethrin, pirimiphos-methyl and control). Within each exposure interval, means followed by the same lowercase letter did not differ significantly according to the Tukey–Kramer HSD test at $p < 0.05$. Where no letters exist, no significant differences were noted. ANOVA parameters were noted for Day 3 (F=243.02, $p < 0.001$); for Day 7 (F = 1209.4, $p < 0.001$); for Day 14 (F = 342.77, $p < 0.001$) and for Day 21 (F = 198.05, $p < 0.001$). On Day 3, data were exp. transformed and the Brown–Forsythe test was utilized (F = 1.01, $p = 0.398$); on Day 7, data were Log10-transformed and the Brown–Forsythe test was utilized (F = 1.77, $p = 0.174$); on Day 14, the O'Brien test was utilized ($p = 0.304$); and on Day 21, data were Lgamma-transformed and the O'Brien test was utilized (F = 2.79, $p = 0.055$).

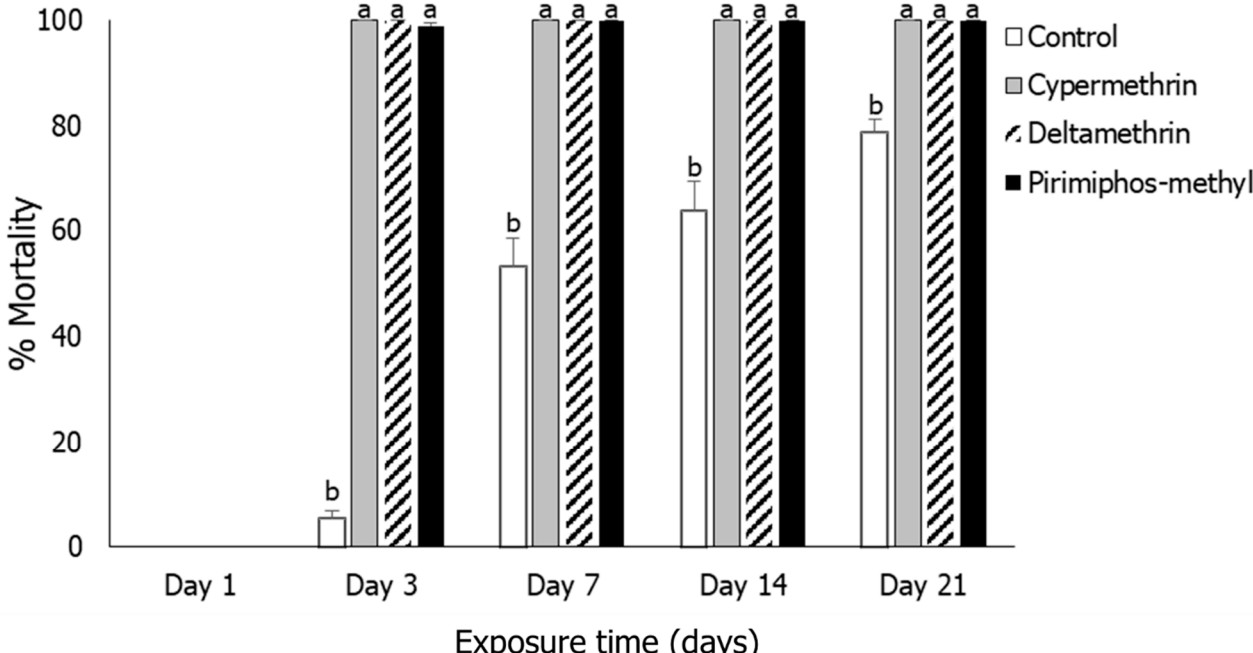

**Figure 7.** Mortality means (% ± SE) of *Oryzaephilus surinamensis* larvae after exposure for 1, 3, 7, 14 and 21 days to rice treated with three insecticides (cypermethrin, deltamethrin, pirimiphos-methyl

and control). Within each exposure interval, means followed by the same lowercase letter did not differ significantly according to the Tukey–Kramer HSD test at $p < 0.05$. Where no letters exist, no significant differences were noted. ANOVA parameters were noted for Day 3 (F = 3018.66, $p < 0.001$); for Day 7 (F = 80.41, $p < 0.001$); for Day 14 (F = 41.01, $p < 0.001$) and for Day 21 (F = 82.51, $p < 0.001$). On Day 3, data were exp. transformed and the Brown–Forsythe test was utilized (F = 2.28, $p = 0.09$); on Day 7, data were Log10-transformed and the O'Brien test was utilized (F = 2.65, $p = 0.064$); on Day 14, the O'Brien test was utilized ($p = 0.119$); and on Day 21, data were exp. transformed and the O'Brien test was utilized (F = 1.13 and $p = 0.348$).

## 4. Discussion

The results of the current study demonstrate that certain insecticides that are currently in use as grain protectants can effectively control a wide range of primary and secondary colonizers in rice fortified with basil. Given that there are no data available for this specific commodity (rice fortified with basil), we provide evidence that there are some factors that can be further taken into account in this field of study, considering the earlier study by Agrafioti et al. [38]. In previous research, the effect of basil on the progeny production of stored-product insects on basil-fortified rice was underlined; in some of the cases examined, it turned out to be detrimental [38]. In some of the tested combinations, there were some differences between the application of spearmint and that of basil [38]. The results of that study are certainly encouraging for the further utilization of the characteristics of fortified rice for stored-product-insect control.

The insecticides used here had a dissimilar efficacy among the species tested. In this context, *S. oryzae* was by far the most susceptible to diatomaceous earth. This result is in accordance with the earlier findings of Athanassiou et al. [40] that illustrated that a commercially available diatomaceous earth formulation was able to provide increased mortality for *S. oryzae* adults, but the type of rice was a crucial factor that determined efficacy. In contrast, *R. dominica* and, especially, *T. castaneum* adults were by far the most tolerant to diatomaceous earths among the species tested. Both species are regarded as being tolerant to inert materials as well [32,41,42], despite the fact that there are several studies that show the opposite [43,44]. *Rhyzopertha dominica* adults are slow-moving, which means that their contact with diatomaceous earth particles is rather reduced compared with other species that move faster [13,44]. On the other hand, *T. castaneum* adults, although less slow-moving than those of *R. dominica*, are also tolerant to diatomaceous earths [32]. However, as it is evident from the data of this study as well, the population of *T. castaneum* can be gradually reduced through increased larval mortality, meaning they are considered susceptible to diatomaceous earth [33].

Our bioassays show that *R. dominica* is tolerant to pirimiphos-methyl. While cases of resistance of this species to this insecticide have been reported from several parts of the globe [29,45,46], it seems that this species has a natural tolerance to pirimiphos-methyl [45]. For example, Rumbos et al. [46] highlighted that pirimiphos-methyl was ineffective against *R. dominica* even at elevated dose rates which were lethal to other major stored-product beetle species. Conversely, *R. dominica* was susceptible to both pyrethroids tested, with deltamethrin being slightly more effective than cypermethrin at short exposure intervals. Earlier studies clearly demonstrate the efficacy of deltamethrin against *R. dominica* [15,26,47] and the potential of this insecticide to be utilized successfully against populations which are resistant to other active ingredients. However, there are cases of resistance development to deltamethrin by *R. dominica* [48].

All three insecticides were effective against *S. oryzae*, as complete mortality or mortality close to 100% was recorded even after 7 days of exposure. Nevertheless, as noted in the case of *R. dominica*, pirimiphos-methyl proved to be the least effective insecticide against both *T. castaneum* and *O. surinamensis*, compared with pyrethroids; however, as above, larvae of both species were more susceptible to pirimiphos-methyl than adults, which is expected to gradually decrease adult density in the treated substrate through decreased

immature development. Between the two pyrethroids, deltamethrin was more effective than cypermethrin in most of the combinations examined, especially for *T. castaneum* adults, where cypermethrin was practically ineffective. Earlier works have reported that deltamethrin is highly effective against *T. castaneum* when applied either on the grain or on surfaces [15,49,50]. Specifically, the deltamethrin-emulsifiable formulation was found to be effective on plastic surfaces and when sprayed directly on insects, but was not effective as a residual insecticide on concrete surfaces [50].

**5. Conclusions**

In conclusion, the tested diatomaceous earth was effective against adults of *S. oryzae* and larvae of *O. surinamensis*. In this context, the most susceptible species was *Sitophilus oryzae.* Between the two pyrethroids, deltamethrin was more effective than cypermethrin in the tested insecticides. The rotation of insecticides in stored-product protection is one of the key elements in integrated pest management strategies at the post-harvest stages of durable agricultural commodities. Given that there are no data available for fortified rice, this work confirms the value of a preliminary screening of the insects present in a certain location that is to be treated before making wider decisions on how to proceed with specific insecticides. Apart from stored grain protection, the use of more than one active ingredient can result in additive effects at lower concentrations, minimizing the risks of residues in the final commodity.

**Author Contributions:** Conceptualization, C.G.A. and P.A.; methodology, P.A., M.V. and C.I.R.; investigation, M.V. and P.A.; resources, C.G.A.; data curation, P.A.; writing—original draft preparation, P.A., E.L. and C.I.R.; writing—review and editing, P.A., E.L., C.I.R. and C.G.A.; visualization, P.A. and M.V.; supervision, C.G.A.; project administration, C.G.A.; funding acquisition, C.G.A. All authors have read and agreed to the published version of the manuscript.

**Funding:** This study was co-financed by the European Union and Greek national funds through the Operational Program Competitiveness, Entrepreneurship and Innovation, under the call RESEARCH-CREATE-INNOVATE (project code: T2EΔK-03726).

**Data Availability Statement:** Data are contained within the article.

**Conflicts of Interest:** The authors declare no conflict of interest.

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
