# Peer review of "Efficacy of Four Insecticides Applied to Fortified Rice with Basil against Major Stored-Product Insect Species"

_agronomy, doi:10.3390/agronomy13123055_

Round 1
Reviewer 1 Report
Comments and Suggestions for Authors
Is the “fortified rice with basil” in the title, same as the “basil-fortified rice” in the text.
Line101: larvae of T. castaneum and O. surinamenis were utilized in the bioassays, how about the instar of the larvae, is it the last-instar larvae, as same with the last-instar larvae in Line107.
Line108: insect cultures have been maintained under controlled conditions for more than 20 years. Why use insect cultures maintained for more than 20 years?
Line 111-112: The procedure for the rice fortification was previously de-scribed by Igoumenidis et al. [48] and Agrafioti et al. [50]. It is too simple to know for readers, perhaps it will be easier for readers if it is described based on the reference of [48] and [50].
Line199: S in S oryzae should be S.
Author Response
Manuscript Number: agronomy-2554231
Section: Pest and Disease Management
Special Issue: Post-harvest Pest and Disease Management
Title: Efficacy of four insecticides applied to fortified rice with basil against major stored product insect species 
Dear Editor,
We would like to resubmit the manuscript entitled: “Efficacy of four insecticides applied to fortified rice with basil against major stored product insect species”.
I remain at your disposal should you need additional information.
Many thanks for your interest on our work.
Sincerely,
Paraskevi Agrafioti
Laboratory of Entomology and Agricultural Zoology
University of Thessaly
Comments:
Is the “fortified rice with basil” in the title, same as the “basil-fortified rice” in the text.
REPLY: Yes, it is exactly the same.
Line101: larvae of T. castaneum and O. surinamenis were utilized in the bioassays, how about the instar of the larvae, is it the last-instar larvae, as same with the last-instar larvae in Line107.
REPLY: Correct. For both species, last-instar larvae was tested in the bioassays. Please see line 107.
Line108: insect cultures have been maintained under controlled conditions for more than 20 years. Why use insect cultures maintained for more than 20 years?
REPLY: The insect cultures are renewed every month. All rearings have been maintained in the Laboratory of Entomology and Agricultural Zoology for more than 20 years.
Line 111-112: The procedure for the rice fortification was previously de-scribed by Igoumenidis et al. [48] and Agrafioti et al. [50]. It is too simple to know for readers, perhaps it will be easier for readers if it is described based on the reference of [48] and [50].
REPLY: Revised, please see line 112.
Line199: S in S oryzae should be S.
REPLY: Correct, please see line 199.

Reviewer 2 Report
Comments and Suggestions for Authors
The authors represent a study of protecting stored fortified rice against four major insect species using four commercial insecticides. The topic is original and the results contribute to the new findings of protecting fortified cereal stock against adult and larvae pests. The methodology is well described and the study is designed correctly. Overall, the results are clear and presented concisely. The conclusions are supported by the findings. The cited references are relevant, but only 28% were published within the last 5 years, which could be improved with more recent ones. Some points could be changed (detailed comments and recommendations are listed within the attached file), and after the authors refine the manuscript, it could be appropriate for publication.

Comments on the Quality of English Language
The English language should be improved, there are some misspelling and incorrect use of the prepositions. I recommend that authors use English proofreading.
Author Response
The authors represent a study of protecting stored fortified rice against four major insect species using four commercial insecticides. The topic is original and the results contribute to the new findings of protecting fortified cereal stock against adult and larvae pests. The methodology is well described and the study is designed correctly. Overall, the results are clear and presented concisely. The conclusions are supported by the findings. The cited references are relevant, but only 28% were published within the last 5 years, which could be improved with more recent ones. Some points could be changed (detailed comments and recommendations are listed within the attached file), and after the authors refine the manuscript, it could be appropriate for publication.
REPLY: The authors would like to thank the reviewer for the comments and recommendations on improving the quality of the manuscript before resubmission. The comments are valuable and contribute greatly to the improvement of our work. In the following we shall refer to these comments, which, as you can see have been addressed/incorporated.
COMMENTS AND RECOMMENDATIONS
Abstract
Line 11 Delete comma after foods
REPLY: Revised, see line 11.
Line 12 Change To ensure the… with Ensuring food
REPLY: Revised, see line 12.
Line 15 Change the vast majority of stored…with the major stored…..
REPLY: Revised, see line 15.
Line 18 Change the sentence with: Results of the diatomaceous earth treatments showed that…..
REPLY: Revised, see line 18.
Line 22 Change …in the vast majority of the combinations examined with … in the tested insecticides.
REPLY: Done, see line 22.
Line 23 Delete clearly. Between insecticides and which, add …on the market
REPLY: Done, see lines 23-24.
Introduction
Line 28 Delete as
REPLY: Done, see line 28.
Line 34 Change rice producing with rice-producing
REPLY: Done, see line 34.
Line 35 Change Madagarcar with Madagascar
REPLY: Done, see line 35.
Line 42 Change disinfestation for the most… with disinfestation of most…
REPLY: Revised, see line 42.
Line 49 Miss spelling cypermethrin
REPLY: Done, see line 49.
Line 51 use plural for insect
REPLY: Done, see line 51.
Line 54 Use term several instead of a number of
REPLY: Revised, see line 54.
Line 58 Delete preposition a (…on partially treated rice…)
REPLY: Done, see line 58.
Line 65 Delete preposition the
REPLY: Done, see line 65.
Line 69 Use plural of the word type
REPLY: Done, see line 69.
Line 71 Use plural of the word stage
REPLY: Done, see line 71.
Line 76 Miss spelling Coleoptera
REPLY: Revised, see line 76.
Line 78 Change that with to
REPLY: Done, see line 78.
Line 88 Instead of …both chemicals and physicals use the term …with chemical and physical mode of action
REPLY: Revised, see line 88.
Line 94 Type correct name Coleoptera
REPLY: Done, see line 94.
Material and Methods
Line 100 Use plural of the word stage
REPLY: Revised, see line 100.
Line 101/102 Change aforementioned with tested. Delete all (before maintained)
REPLY: Done, see lines 101-102.
Line 107 Use term: …less than a month old...
REPLY: Revised, see line 107.
Line 110 Use hereafter instead of thereafter
REPLY: Done, see line 110.
Line 112 Put to after Prior
REPLY: Done, see line 112.
Line 113 Put preposition a before controlled
REPLY: Done, see line 113.
Line 115 and 117 Use K-Othrine® and Actellic® 50 EC
REPLY: Done, see lines 115 and 117.
Line 118/119 Start the sentence as: The diatomaceous earth Silicid (…) was used...
REPLY: Revised, see lines 118-119.
Line 125 Start the sentence as: The DE formulation was applied at….
REPLY: Revised, see line 125.
The authors should choose what term they will use for diatomaceous earth, will it be DE or full name, but it must be constant throughout the manuscript.
REPLY: Correct, we use the term “diatomaceous earth” throughout the manuscript.
Line 126 Use prepared instead of utilized
REPLY: Revised, see line 126.
Line 131 Put preposition a in front of control. Change the sentence as: Treated commodity was transferred…
REPLY: Revised, see line 131.
Line 139 Use preposition the for the word insectʾs ecsape
REPLY: Done, see line 139.
Line 142 Miss spelling Individual
REPLY: Revised, see line 142.
Line 144 Change normalization with normality
REPLY: Revised, see line 144.
Results
Line 161-163 Use italic font for the insect species names, and also in the table
REPLY: Done, see lines 161-163.
Line 183 Put preposition the for the statistical test names; the Tukey -Kramer HSD, the OʾBrien test, the Brown-Forsythe ……please change whenever it is mentioned
REPLY: Done.
Line 215 Change the second part of the sentence as: …T. castaneum adults in rice treated with…
REPLY: Revised, see line 215.
Line 220 Delete further. Change ...in treated rice… to … in rice treated with….
REPLY: Done, see line 220.
Line 24 Put preposition the before the vast
REPLY: Done, see line 241.
Line 246 Delete preposition the before deltamethrin
REPLY: Done, see line 246.
Line 262 Change the preposition: instead of at Day 1 use on Day 1, and change whenever it is mentioned. Miss spelling adults. Use plural for the word Figure
REPLY: Revised, see line 262.
Line 264/265 It should be clearly indicated in which treatments was the mortality high, in the treatment with DE or insecticide, or both?
REPLY: Figure 6 is referred to insecticide treatments. We have included the phrase “…. in the treatment with insecticides…..” see lines 264-265.
Line 269/270 put comma after (Figure 6), continue the sentence with… where no significant differences….
REPLY: Revised, please see lines 269-270.
Line 283-287 The mortality of O. surinamensis larvae was too high in the control treatment (>50%, >60% and almost 80% after 7, 14 and 21 d). How do you explain that? Have you calculated corrected efficacy using some of the formulas for corrections?
REPLY: Thank you for your comment. Indeed, the control mortality of O. surinamensis larvae was high, since <80% was recorded after 21 days of exposure. Firstly, sound kernels basil-fortified rice was the commodity used in our bioassays which is not the preferable substrate for O. surinamensis development, since this insect species prefers cracked kernels as a secondary pest (Lampiri et al., 2023).
Although technically classified as an external-feeding pest on grains and cereals that can only infest kernels that have been previously damaged by internal-feeding pests (Mahroof and Hagstrum 2012, Suleiman and Rugumamu 2017), there are occasional reports that this pest can also attack intact kernels, but less successfully in comparison with other major stored product insect species (Hill 2003).
Please see some references below:
- Mahroof, R. M., and D. W. Hagstrum. 2012. Biology, behavior, and ecology of insects in processed commodities, pp. 33–44. In D. W. Hagstrum, T.W. Phillips, and G. Cuperus (eds.). Stored product protection. Kansas State University, Manhattan, KS.
- Suleiman, M., and C. P. Rugumamu. 2017. Management of insect pests of stored sorghum using botanicals in Nigerian traditional stores. J. Stored Prod. Postharvest Res. 8: 93–102.
- Hill, D. S. 2003. Pests of storage foodstuffs and their control. Kluwer Academic Publishers, New York.
- Lampiri, E., Scully, E.D., Arthur, F.H., Athanassiou, C.G. 2023. Development and immature mortality of the sawtoothed grain beetle (Coleoptera: Silvanidae), on different sorghum fractions and different temperatures. J. Econ. Entomol. 116, 615-620.
Discussion
Line 302 Change …there are…with …there is….
REPLY: Revised, see line 302.
Line 305/306 Change the sentence as: In previous research the effect of basil has been underlined on progeny production of stored product insects on basil-fortified rice, turned out to be detrimental in some of the cases examined.
REPLY: Revised, see lines 305-306.
Line 314 Put as after regarded
REPLY: Done, see line 314.
Line 318 Change One… with On…
REPLY: Revised, see line 318.
Line 321 Change …which are… with …which is…
REPLY: Done, see line 321.
Line 326 Change …non effective… with …ineffective…
REPLY: Done, see line 326.
Line 327 Change …that… with …which…
REPLY: Done, see line 327.
Line 329 Miss spelling cypermethrin
REPLY: Done, see lline 329.
Line 321 Delete…. ,which are….
REPLY: Done, see line 321.
Line 334 Change …effectively managed to control S. oryzae… with …effective against S. oryzae …
REPLY: Revised, see line 334.
Line 340 Delete …it seems that….
REPLY: Done, see line 340.
Line 345 Put preposition the before rotation
REPLY: Done, see line 345.
Line 347 Change …there are no data… with …there is no data…
REPLY: Done, see line 347.
Line 348 Delete ...that are…
REPLY: Done, see line 348.
Line 350 Use singular of ingredients
REPLY: Done, see line 350.

Reviewer 3 Report
Comments and Suggestions for Authors
It is difficult to follow the results, in terms of the effects of insecticides and diatomaceous earth. I suggest a figure showing all results if possible. Discussion may be improved.
Author Response
It is difficult to follow the results, in terms of the effects of insecticides and diatomaceous earth. I suggest a figure showing all results if possible. Discussion may be improved.
REPLY: The authors would like to thank the reviewer for the comment on improving the quality of the manuscript before resubmission. Regarding the results section, the tested species are presented in each paragraph and linked to each figure. At the beginning of each paragraph, diatomaceous earth formulation is presented (Figure 1). In the discussion part, some modifications have been completed.

Reviewer 4 Report
Comments and Suggestions for Authors
Overall, the work is well presented and well structured. The manuscript is clear and reasonably relevant to the topic of the study. The document has a good number of citations, but most of them (72%) are not recent publications (within the last 5 years). The document has a small number of self-citations. The experimental design used is appropriate for testing the hypothesis. The results of the manuscript are reproducible as indicated in the methods section, easy to interpret and understand, and adequately and consistently capture the data throughout the manuscript. The figures are generally appropriate. They are easy to understand and adequately and consistently reflect the data throughout the manuscript.
· In lines 90 and 92 use the third person, avoid using we performed and change it to was performed.
· In lines 157, 165, 199, 215, and 261, numbering should follow the previous digit, e.g., 3,1, 3,2, etc.
· It would be interesting to discuss the possible biological and metabolic reasons for the susceptibility and resistance of some insects to the insecticides tested. For example, line 203 states that, curiously, mortality rates increased after three days of exposure for all insecticide formulations tested, but does not discuss the possible reasons for this.
· If possible, include conclusions of the paper highlighting the key findings.
Author Response
Overall, the work is well presented and well structured. The manuscript is clear and reasonably relevant to the topic of the study. The document has a good number of citations, but most of them (72%) are not recent publications (within the last 5 years). The document has a small number of self-citations. The experimental design used is appropriate for testing the hypothesis. The results of the manuscript are reproducible as indicated in the methods section, easy to interpret and understand, and adequately and consistently capture the data throughout the manuscript. The figures are generally appropriate. They are easy to understand and adequately and consistently reflect the data throughout the manuscript.
REPLY: The authors would like to thank the reviewer for the comments and recommendations on improving the quality of the manuscript before resubmission. The comments are valuable and contribute greatly to the improvement of our work. In the following we shall refer to these comments, which, as you can see have been addressed/incorporated.
In lines 90 and 92 use the third person, avoid using we performed and change it to was performed.
REPLY: Revised, third person was used throughout the text.
In lines 157, 165, 199, 215, and 261, numbering should follow the previous digit, e.g., 3,1, 3,2, etc.
REPLY: Done, please see line 165, 199, 215, 261. We did not include the sub-section “insect mortality” since it is the introductory part of the results.
It would be interesting to discuss the possible biological and metabolic reasons for the susceptibility and resistance of some insects to the insecticides tested. For example, line 203 states that, curiously, mortality rates increased after three days of exposure for all insecticide formulations tested, but does not discuss the possible reasons for this.
REPLY: We have included biological reasons for the susceptibility of R. dominica in the discussion part. Please see lines 317-323.
If possible, include conclusions of the paper highlighting the key findings.
REPLY: Done, please see the last paragraph of the manuscript.
